# Experimental Investigation of Polymer-Coated Silica Nanoparticles for Enhanced Oil Recovery

**DOI:** 10.3390/nano9060822

**Published:** 2019-05-31

**Authors:** Alberto Bila, Jan Åge Stensen, Ole Torsæter

**Affiliations:** 1PoreLab Research Centre, Department of Geoscience and Petroleum, Norwegian University of Science and Technology (NTNU), S. P. Andersens veg 15a, 7031 Trondheim, Norway; jan.stensen@ntnu.no (J.Å.S.); ole.torsater@ntnu.no (O.T.); 2SINTEF Industry, S. P. Andersens veg 15a, 7031 Trondheim, Norway

**Keywords:** nanoparticles, nanofluid-flood, enhanced oil recovery, interfacial tension, wettability alteration

## Abstract

Recently, polymer-coated nanoparticles were proposed for enhanced oil recovery (EOR) due to their improved properties such as solubility, stability, stabilization of emulsions and low particle retention on the rock surface. This work investigated the potential of various polymer-coated silica nanoparticles (PSiNPs) as additives to the injection seawater for oil recovery. Secondary and tertiary core flooding experiments were carried out with neutral-wet Berea sandstone at ambient conditions. Oil recovery parameters of nanoparticles such as interfacial tension (IFT) reduction, wettability alteration and log-jamming effect were investigated. Crude oil from the North Sea field was used. The concentrated solutions of PSiNPs were diluted to 0.1 wt % in synthetic seawater. Experimental results show that PSiNPs can improve water flood oil recovery efficiency. Secondary recoveries of nanofluid ranged from 60% to 72% of original oil in place (OOIP) compared to 56% OOIP achieved by reference water flood. In tertiary recovery mode, the incremental oil recovery varied from 2.6% to 5.2% OOIP. The IFT between oil and water was reduced in the presence of PSiNPs from 10.6 to 2.5–6.8 mN/m, which had minor effect on EOR. Permeability measurements indicated negligible particle retention within the core, consistent with the low differential pressure observed throughout nanofluid flooding. Amott–Harvey tests indicated wettability alteration from neutral- to water-wet condition. The overall findings suggest that PSiNPs have more potential as secondary EOR agents than tertiary agents, and the main recovery mechanism was found to be wettability alteration.

## 1. Introduction

The production rates of existing oil fields are declining and the frequency of new explorations has become scarce. Therefore, the significance of enhanced oil recovery (EOR) techniques is highly understood by oil companies [1,2]. In this regard, nanoparticles (NPs) have been researched as additives to improve water flooding oil recovery. The small size (1–100 nm) of the nanoparticles and large surface area-to-volume ratio increase the particle mobility and surface activity, particularly at elevated temperature, which contribute to alter the fluid–rock interface properties [2,3,4]. This makes the NPs appropriate for oil production. Moreover, NPs can travel long distances and reach untouched zones of a reservoir with no severe impact to permeability [3,5,6]. The expected outcome is the mobilization of capillary trapped-oil or by-passed oil during the early stages of oil production, thus improving the microscopic sweep efficiency of water flooding. Thus far, silica NPs are the most studied nano-materials for EOR-applications [7,8,9,10], as they are the most abundant compounds on Earth and are of natural occurrence in sandstone formations. More importantly, silica NPs can be easily produced and surface modified to target specific applications [3,11].

Previous studies have reported promising oil recovery results due to silica NPs injection [5,6,12,13,14,15,16,17,18,19,20] under a variety of laboratory conditions. The flooding experiments varied from the use of silica NPs with different size, composition, surface modification, etc., and the NPs suspended in aqueous solutions with diverse ionic strength [5,6,12,13,14,15,16,17,18,20,21,22] or in non-aqueous solutions [12]. The concentration of silica NPs in the suspension is also varied. Despite the potential of the nanoparticles, there is a lack of experimental repeatability. This poses many challenges in assessing the effectiveness of silica NPs for EOR applications and an adequate understanding of the oil recovery mechnisms. The EOR application of NPs is also hampered by poor dispersibility or stability in the injection fluid (e.g., water). Nanoparticles have an immense surface free energy triggered by their ultra-small size and large surface area, which increases the tendency of the NPs to aggregate/agglomerate in solution in an attempt to gain the low energy state [23]. As the NPs approach each other, their surface functionalities at the fluid–rock interfaces are reduced. Additionally, the binding of the primary NPs to form large nano-structures can block reservoir pore channels [24] from start of the injection, and increase the pressure and costs of oil production. To overcome the stability issues, the surface of the nanoparticles can be modified by attaching or coating macro-molecules (e.g., polymers) on NPs surface on it, or new functionalities (silane coupling agents) may be given to the surface of the NPs to prevent them from attracting each other or with the surrounding molecules.

Nanoparticles with polymer chains coating on the particle surface were recently proposed for EOR applications. These nanomaterials are often referred to as polymer-coated NPs, and they can offer significant advantages such as improved solubility and stability, greater stabilization of emulsions, and easier transport through porous media [25,26] over bare nanoparticles. There are still very few experimental studies concerning their EOR applications. Accordingly, the oil recovery mechanisms of the NPs need to be well understood. Recently, Choi et al. [27] demonstrated that polymer-coated silica NPs could enhance oil recovery by 5% volume while lowering the injection pressure. They attributed the EOR mechanisms to the reduced oil and water IFT and the formation of wedge film between the oil and the rock surface that readily altered the wettability. The effect of surface-modified silica NPs to decrease the water injection pressure for EOR purposes was studied by Dai et al. [23] and Zhao et al. [28]. The authors observed that, during the injection of nanofluid, NPs adsorbed and formed layers on the rock surface, thereby crowding the water out on it and increasing the water flowing channels, thus decreasing the injection pressure. The formed layer of NPs also altered the surface wettability. Behzadi and Mohammadi [29] evaluated the EOR potential of polymer-coated silica NPs as secondary EOR-agents using glass micromodels. They reported about 59% oil recovery at breakthrough point compared to 39% OOIP achieved by water flood. This study highlighted the change in wettability of an oil-wet glass towards water-wet as the governing EOR mechanism over oil–water IFT reduction (from 28.2 mN/m to 14.1 mN/m). Azarshin et al. [30] found that contact angle of an oil-wet surface decreased more in the presence of modified-silica NPs than bare silica NPs. This effect contributed for 18% increase in total oil recovery compared to bare NPs in their work. Again, the adhesion of NPs and subsequent formation of nanotexture coating on the rock surface and the reduction of oil–water IFT were invoked as the main oil recovery mechanisms, although the reported reduction of IFT was from 37 mN/m to 12 mN/m. Other experimental studies focusing on the evaluation of modified silica NPs to improve oil recovery have found that the NPs are more efficient in displacing light oils than heavy oils [7,31]. In these studies, the reduction of oil and water IFT from 26.5 to 1.9 mN/m and the change in wettability from water- to oil-wet condition are also identified as the key parameters for oil production.

The formation of such NP layers or nanotextures on the rock surface was experimentally observed by Wasan and Nikolov [32], and the authors proposed the concept of structural disjoining pressure to explain the influence of NPs on rock wettability and oil recovery. The authors asserted that the well-ordered layers formed in the confined space of the wetting wedge, as shown in Figure 1, result in: (i) increased NP concentration in the wedge than in the bulk suspension, which creates an osmotic pressure that attempts to separate the oil from the rock surface, thus increasing oil recovery; and (ii) as the film tension increases toward the vertex of the wedge, an extra force is also developed that promotes the spreading of NPs on the rock surface and wetting effect. Dai et al. [33] added that the spreading of NPs on the rock surface may develop new surface roughness that can change the wettability. It is worth noting that for the structural disjoining pressure mechanism to be effective for EOR, it requires a long-range force or high NP concentration [32,34]. Additionally, there should be no tendency for the NPs to adsorb on the rock surface [35], so that NPs can confine themselves in a pre-existing wetting-wedge [36]. Thus, it seems that the concept of structural disjoining pressure mechanism cannot be generalized to explain the wetting and oil recovery by virtue of nanofluids. For instance, in oil-wet reservoirs, oil films coat on the rock surface, which is unfavorable to the existence of a wetting-wedge.

Thus far, the wettability alteration and the reduction of oil–water IFT are put forth as the main EOR mechanisms of surface-modified silica NPs. It should, however, be noted that the reported IFT reduction in the literature is still modest and cannot prove a surfactant-like behavior of the NPs, since properly chosen surfactants would lower the IFT values down to <10${}^{-3}$ mN/m and contribute significantly to the recovery of oil. This may be the reason the oil recovery due to NPs was attributed to a joint contribution of wettability alteration and reduction of the interfacial tension. The objective of this work was to evaluate the effect of polymer-coated silica NPs for enhancing oil recovery on neutral-wet Berea sandstone rocks and propose possible EOR mechanisms. The NPs were prepared in seawater at a concentration of 0.1 wt %. Two-phase flow experiments were performed directly and after water flood at room temperature. Fluid–rock interface interactions and migration behavior of the NPs through the rock were studied to understand the effect of NPs on oil recovery.

## 2. Experimental Materials and Methods

### 2.1. Polymer-Coated Silica Nanoparticles and Synthetic Seawater

Five types of hydrophilic silica nanoparticles (NPs) were used in this work. These nanomaterials are special research and development laboratory products from Evonik Industries. The particles are spherical and amorphous silica products marketed under AEROSIL^®^ trade name. The original nanoparticles are commercially available from Evonik Industries, and are unstable in synthetic seawater containing divalent cations such as Ca${}^{2+}$ and Mg${}^{2+}$ [37]. Therefore, the surface of the particles was modified by attaching polymer chains on it to provide a long-term stability and supplied to us as AERODISP^®^, which is AEROSIL^®^ particles in liquid solution as shown in Figure 2. The concentration of the particles in distilled water varied from 21.6 wt % to 36.8 wt %. The main component of the particles was silicon dioxide, other components comprised of aluminum oxide, mixed oxides (MOX), etc. These types of nanomaterials are referred to as polymer-coated silica nanoparticle (PSiNP) hereafter. The specific surface area of PSiNPs varied from 140 to 220 m${}^{2}$/g. The selected properties of the NPs are summarized in Table 1. The primary size of the NPs was measured with dynamic light scattering technique. The NF stands for nanofluid and the number is used to identify the NP type.

The concentrated solutions of PSiNPs were diluted at a concentration of 0.1 wt % in SSW. The prepared nanofluid (NF) solution was stirred for approximately 30 min using a magnetic stirrer to ensure a homogeneous solution before use. The SSW was prepared by adding salt components found in North Sea water. The total dissolved salts was 38,318 ppm. The properties of the prepared solutions of SSW and NF are reported in Table 2. The density was measured with Anton Paar density meter DMA^TM^ 4100 M series. A rotating viscometer (Brook-field, model LVDV-II+P) was used to obtain viscosity. The pH of the solutions was measured using a pH Meter (model pH 1000 L, phenomenal^®^).

### 2.2. Oleic Phase

Two types of crude oils were used in this work and were obtained from North Sea fields, here referred to as A and B. The crude oils were filtered twice through a 5 μm Millipore filter. The density and viscosity were measured using the same instruments as in Section 2.1. The measured values are presented in Table 3, including the saturates, aromates, resins, and asphaltenes (SARA) analysis. Crude oil A was used for aging the cores and crude oil B used for additional tests.

Normal decane (dyed red) with density of 0.73 g/cm${}^{3}$ and a viscosity of 0.9 cP was used for wettability experiments.

### 2.3. Interfacial Tension and Amott–Harvey Wettability Tests

Interfacial tension was measured between crude oil B and SSW or nanofluid using pendant drop method at ambient conditions. The Krüss drop shape analyzer (100) was assembled with a J-shape syringe needle with an inner diameter of 1.0047 mm to dose crude oil drops in the bulk phase. The dosing rate was 2.67 μL/s. With the oil drop hanging from the needle in the designated fluid or bulk phase, the measurements were taken every 20 s until the equilibrium was reached (i.e., for ≈2 h). Then, the static IFT value was reported.

The rock wettability was evaluated before and after nanofluid flooding using Amott-Harvey method. This method is one of the most used for the characterization of the wettability of reservoir cores in petroleum engineering. More importantly, it combines spontaneous and forced displacements of the fluids to obtain the average wettability of a core [38]. The main problem with Amott–Harvey method is its insensitivity near neutral-wettability [38]. In this work, we used the following Amott–Harvey evaluation procedure:The core set at irreducible water saturation (S${}_{wir}$) was submerged in an Amott–Harvey cell filled with SSW. The SSW was allowed to imbibe spontaneously into the core displacing oil until equilibrium was reached. The amount of oil produced overtime was step-wise recorded, and the total denoted as V${}_{o1}$.The remaining mobile oil in the core was forcibly produced using a centrifuge down to residual oil saturation (S${}_{or}$). The oil produced at this stage was recorded as V${}_{o2}$.The core at S${}_{or}$ was placed in an Amott–Harvey cell filled with oil. Then, the oil was allowed to imbibe into the core to displace the SSW until the equilibrium was achieved. The total volume of SSW produced was recorded as V${}_{w1}$.The remaining mobile SSW in the core was force produced using a centrifuge and the production recorded as V${}_{w2}$.

Each cycle of spontaneous water and oil imbibition tests were conducted for 15 days, while forced displacement tests were performed using a centrifuge at 5000 rpm and 25 °C for 2 h. Normal decane was used as the oil phase. All experiments were carried out at ambient conditions. The Amott–Harvey wettability index (WI) is the difference between the displacement-by-water ratio (water index, I${}_{w}$ = V${}_{o_{1}}$/(V${}_{o_{1}}+V_{o_{2}}$) and the displacement-by-oil ratio (oil index, I${}_{o}$ = V${}_{w_{1}}$/(V${}_{w_{1}}+V_{w_{2}}$). A categorization of the rock wettability based on WI was given by Anderson [38].

### 2.4. Porous Medium and Preparation Procedure

Twenty-four Berea sandstone core plugs initially at water-wet conditions were used throughout this work. The cores were drilled from the same block of Berea sandstone; the bulk mineral composition was measured by X-ray diffraction. The cores were nearly homogeneous and composed of 93.7 vol % quartz, 5.0 vol % of microcline (alkali feldspar), and diopside (1.3 vol %). The cores were prepared to have similar dimensions in length: ≈4.5 cm, and diameter: ≈3.8 cm. The average porosity of the cores was 18% and permeability ranged from 228–391 mD.

***Cleaning:*** The core plugs were rinsed with methanol through Soxhlet apparatus for approximately 8 h. Then, they were dried at 60 ${}^{\circ }$C for 2–3 days in the oven.***Porosity and Permeability:*** Porosity and permeability were measured on cleaned and dried cores using saturation and helium porosimeter methods and gas permeameter, respectively.*Core saturation:* The cores were vacuumed for 2 h under 100 mbar pressure before allowing SSW to enter the cores. The system was left under vacuum for additional 2–3 h to ensure 100% SSW saturation. Then, they were left submerged in SSW for at least 10 days to establish ionic equilibrium with the rock system.***Irreducible water saturation:*** Centrifuge was used to establish S${}_{wir}$. The cores saturated with SSW were centrifuged in crude oil B at 5000 rpm for 2 h at 25 ${}^{\circ }$C. This procedure also established OOIP;***Core Aging:*** To rupture the water films to allow the polar components of crude oil to adsorb onto rock surface, the original water-wet cores at S${}_{wir}$ were submerged in crude oil A inside metallic containers (aging cells) and placed in an oven set at 80 ${}^{\circ }$C for 4–6 weeks.

### 2.5. Rock Core Flooding

The effect of the PSiNP based nanofluids on oil recovery was evaluated using unsteady-state core flooding procedure in secondary and tertiary oil recovery modes under nearly atmospheric conditions. The flooding tests were performed on induced neutral-wet cores (aged cores). Twin core plugs were used for each nanofluid test. This procedure aimed at reducing the experimental errors and reproduce the results. Prior to the flooding experiments, the aged cores were cooled to room temperature followed by injection of 1–2 pore volumes (PVs) of fresh crude oil B. This was to eliminate air bubbles and/or re-establish oil connectivity within the cores. Secondary nanofluid flooding was then performed at 0.2 mL/min until no oil production was occurring for 2–4 PVs. Afterwards, the injection was switched to water flooding (WF) at 0.5 mL/min to measure if additional oil could be recovered with the increased flow-rate.

In the case of tertiary flooding scheme, WF was conducted before nanofluid flooding under the same flow-rate of 0.2 mL/min. However, the injection was switched from water to nanofluid flooding after no oil was produced for 1–2 PVs. This procedure was to ensure that any oil produced afterwards was a result of nanofluid effect. During the experiments, the amount of oil produced was collected every 14 PV and corrected for the flooding system dead volume. When the oil production was produced at low pace, a camera with automated capturing was set to record the oil production in a graded line overtime while the total production was being collected in a larger graded effluent collector. The pictures were then analyzed to determine the amount of oil production overtime. This oil was compared to the total oil production volume read off in the large graded effluent collector. Meanwhile, the oil recovery factor (RF) and differential pressure (dP) across the core were recorded versus PVs, and the residual oil saturation was calculated.

Figure 3 presents a schematic of core flooding experimental setup with its main components labeled. It utilizes an injection pump, three tanks containing crude oil, SSW and nanofluid each. All tanks were assembled vertically inside a temperature controlled oven. The core was loaded in the Hassler core-holder and oriented horizontally under confining pressure held within 18–22 bar. Two pressure sensors were connected at the inlet and outlet of the core-holder to monitor the fluid differential pressure (dP) across the core. The differential pressure was recorded every 15 s. The outlet of the core was at atmospheric pressure as no back pressure was used.

## 3. Results

### 3.1. Aging

Six core plugs initially water-wet were aged in crude oil A (at S${}_{wir}$) using the procedure described in Section 2.4. After aging was completed, the crude oil within the core was displaced by centrifuging the core in decane. Afterwards, the Amott–Harvey wettability test was performed. Table 4 presents the measured WIs data. It can be seen that almost no SSW imbibed spontaneously into the cores, thus the Amott water index, I${}_{w}$, is zero for all cores. On the other hand, small amounts of oil imbibed into the cores; the Amott oil index, I${}_{o}$, varied from 0.09 to 0.17. The average Amott–Harvey WIs was −0.1 and it varied from −0.07 to −0.17. The overall results indicate that the wettabilities of the cores were altered from strongly water-wet to neutral-wet condition with aging process. It is worth mentioning that these cores were only used to evaluate the efficiency of aging process. Additional core plugs were prepared and aged under the same procedure, assuming that it would give the same results.

### 3.2. Core Flooding

#### 3.2.1. Secondary Nanofluid Flooding

Eight core flooding tests were conducted in this section to evaluate the EOR potential of the PSiNPs. Two parallel flooding tests were performed for each nanofluid. The average oil recoveries due to nanofluid flooding varied from 59.3% to 71.5% of OOIP compared to 56.05% OOIP on average from reference WF conducted under the same conditions. The main oil was produced before the breakthrough point (BT), but there were additional oil production as the injection advanced, which revealed good EOR-potential of the nanofluid systems. Table 5 summarizes the main core flooding parameters used to evaluate the performance of the nanofluids.

The average pore volume (PV) injected during water flood to reach the maximum recovery was about 3, while the nanofluid injection needed approximately twice of this amount (PVs injected for water flood). Due to the continual microscopic sweep efficiency of the nanofluids, the oil recovered was higher than reference water flood. In the secondary recovery mode, the nanofluids increased oil recovery by factors of 3.3 to 15.5% points compare with water flood. When the injection was switched from nanofluid flood to WF and the flow-rate increased, additional oil was recovered and it ranged from 0.66 to 2.15% of OOIP.

The differential pressure (dP) across the core due to nanofluid injection initially increased for approximately 0.5 PVs, then it decreased and it remained lower than the maximum WF differential pressure under the same injection procedure. The exception was the injection of NF3; where the dP gradually increased for 1.3 PVs and then it seemed to level. With continued injection, the pressure was spiky and the maximum dP reached at the end of the flood test was higher than that achieved by the reference WF. Figure 4 is an example used to illustrate the behavior of dP and the oil recovery plotted against PVs injected for samples NF3 and NF5. The differences in pressure behavior for the same nanofluid flooding can be attributed to core properties.

Similar plots of dP and oil recovery profiles for samples NF1 and NF2 are presented in Appendix A (Figure A1a,b). At this stage, NF5 had the largest oil recovery, while showing good ability to decrease the dP (Figure 4b).

#### 3.2.2. Tertiary Nanofluid Flooding

In this section, ten core flooding tests were performed (two parallel tests for each nanofluid). This included sample NF4, because it was believed it would give better results than NF3. These two samples had similar particle composition and surface modification. The main difference between the two samples was the primary particle size and particle concentration in distilled water as provided by manufacturer in Table 1. Water flooding was conducted prior to nanofluid flooding. Both tests were conducted at the same flow-rate of 0.2 mL/min. The oil displacement efficiency (E${}_{D}$) due to nanofluid was evaluated using the following equation:(1)$$E_{D}=\left[1-(\frac{S_{or2}}{S_{or1}})\right]x100\%$$ where S${}_{or1}$ and S${}_{or2}$ represent residual oil saturation at end of WF and nanofluid flooding, respectively.

The average oil recovery achieved by WF was 56% (±4.24) of OOIP and it ranged from 50% to 61% of OOIP. The incremental oil recovery achieved by nanofluid flooding ranged from 2.6% to 5.2% OOIP. The corresponding displacement efficiency varied from 6.3% to 11.8%. The oil recoveries and the main parameters used to assess the viability of the nanofluids are given in Table 6. All nanofluids could mobilize to some extent the residual oil, thus increasing oil recovery. It can be seen in Table 5 that there was no significant oil recovery within the first pore volumes. The ultimate incremental recoveries were achieved after large volumes of nanofluids were injected. Selected plots of oil recovery and differential pressure versus PVs injected are illustrated in Figure 5. Similar plots are provided in Figure A1 for the remaining nanofluid samples. Figure 5a shows that the dP was decreased just after the injection of nanofluid NF2. Similar pressure patterns were observed for NF1 and NF5. By contrast, the pressure increased with the injection of samples NF3 and NF4 indicating that some pore plugging or log-jamming events were happening during the oil displacement process. An example of this pressure pattern is illustrated in Figure 5b. The squares in Figure 3a,b show the pressure decreasing effect and zones of jammed {NP}s during the nanofluid injection, respectively. 

The twin cores used for each nanofluid flood gave small variations in oil recoveries indicating reproducibility of experimental results, i.e., <5%. The recovery from samples NF3 and NF4 indicated positive correlation with increased dP (i.e., increased oil recovery with increasing injection pressure), while, among the samples exhibiting the pressure-decrease effect, no clear correlation was found.

### 3.3. Interfacial Tension Results

The interfacial tension (IFT) between crude oil and each nanofluid with a concentration of 0.1 wt % was measured values were measured using pendant drop method. All tests were conducted at ambient conditions until stable value of IFT was reached. It was observed that the tension between crude oil and synthetic seawater (SSW) was decreased in the presence of polymer-coated silica nanoparticles. Figure 6 shows that approximately 2 h were sufficient to reached stable IFT values. The IFT between crude oil and SSW was 10.6 mN/m. This value was decreased to a range of 6.8 to 2.5 with nanoparticles. The largest reduction of IFT was achieved by NF5 mN/m while NF3 showed the lowest reduction to a value of 6.8 mN/m.

### 3.4. Measurements after Core Flooding

#### 3.4.1. Core Wettability

After the core flooding was completed, one of the two cores was removed from the core holder and soaked in the injected nanofluid for 10 days at 40 ${}^{\circ }$C, while the second core was immediately submitted for wettability evaluation. By soaking the cores, the goal was to evaluate the effect of prolonged exposure of PSiNPs on the rock surface. The nanofluid flooded core was centrifuged in decane down to S${}_{wir}$, after which the cores were placed in the Amott–Harvey cells. It was observed that SSW rapidly imbibed into the core displacing significant amount of oil. The oil was recovered from all the faces of the cores from start. Figure 7 shows the spontaneous imbibition behavior curves for both soaked and non-soaked cores that were initially flooded with NF2 and NF3. Other curves are shown in Figure A3 in Appendix B. The SI tests were conducted for 15 days at ambient conditions.

After spontaneous and forced imbibition cycles in water and oil, respectively, the WIs of the cores were determined. The average WI calculated after secondary nanofluid injection (Section 3.2.1) was 0.69 (±0.04) and it ranged from 0.65 to 0.76 (for soaked cores), while for non-soaked cores it was 0.59 (±0.10) and varied from 0.43 to 0.71. When the nanofluid was injected after WF (Section 3.2.2), the average WI was 0.58 (±0.07) and it varied from 0.46 to 0.66 (for soaked cores). For non-soaked cores, it was 0.61 (±0.08) and ranged from 0.51 to 0.73. The results of rate of SI and WIs suggested that the wettability was altered from neutral- to water-wet condition, regardless of the NPs exposition time on the rock surface.

#### 3.4.2. Core Permeability

The core absolute permeability was measured before and after nanofluid flooding. The cores (at S${}_{or}$) were cleaned with toluene and methanol for several days and then dried. This measurement was conducted to determine if there would be changes inside the core resulting from NPs adsorption or retention during the nanofluid flooding. The results of permeability are presented in Table 7. The negative values indicate permeability impairment, whereas the positive ones suggest an improvement.

The absolute permeability measured after secondary injection indicated core impairment with all nanofluids. After tertiary injection, the permeability varied from negative to positive values, indicating both impairment and improvement, respectively. Variable results were observed in cores injected with NF1, NF2 and NF5 as tertiary EOR agents. The core injected with NF1 showed that the permeability was improved. Unfortunately, the second test failed and the results could not be reproduced. The cores injected with NF2 and NF5 exhibited both results (permeability reduction and improvement). The injection of NF3 and NF4 caused the highest core permeability impairment, which is consistent with increased pressure during the injection in both the schemes and the large particle size. The effect of samples NF1, NF2 and NF5 on core permeability was less pronounced compared to NF3 and NF4. There was no visible correlation between the core permeability measured after injecting samples NF1, NF2 and NF5 with initial particle size or with the pressure behavior.

#### 3.4.3. Nanoparticle Size Distribution

The particle size distribution was characterized to aid the stability analysis of the nanofluids. The size of the NPs was measured four months after the nanofluids were prepared and it was compared to that provided by the manufacturer. Additional particle size measurements were performed on the effluent solution to infer whether particle aggregation/agglomeration occurrf or not within the core plugs. The main uncertainty with the effluent analysis may be due to by the rock dissolution and the detachment of the sand grains, thus not providing reliable particle size characterization. The effluent was collected at the very end of the flooding experiment (when oil had ceased production) and the particle size characterized within 24 h using dynamic light scattering technique. The results are presented in Table 8. The diameter of the NPs is reported on average of three measurements conducted for each particle type in SSW at 0.1 wt %. The average primary particle diameter provided by the manufacturer is also included. It was measured with the NPs suspended in distilled water with concentration varying from 21.6 wt % to 36.8 wt % (Table 1) with dynamic light scattering method.

The particle size distribution profile by intensity is exemplified graphically in Figure 8, for sample NF1. The size distribution by intensity describes the contribution of each particle or individual peaks in the distribution. The z-average displayed on the distribution curves is the intensity overall mean size of the particles and it is referred to as particle diameter size. Three measurements were performed in this work, and the average dominant particle size in synthetic seawater is reported in Table 8. The remaining curves are presented Figure A4, Figure A5, Figure A6 and Figure A7 (in Appendix C).

## 4. Discussion

### 4.1. Oil Recovery

To evaluate how effective the polymer-coated silica nanoparticles (PSiNPs) are for enhancing oil recovery, we used both secondary and tertiary injection modes. Two nearly identical core plugs were used for each nanofluid type to measure the reproducibility of the results. Secondary nanofluid flooding results were compared to reference SSW flood. Emphasis was also placed on the oil recovery factor (RF) at the BT point, and the PVs injected to achieve ultimate oil recovery. In tertiary flooding scheme, the arrival of first oil production due to nanofluid was systematically monitored to determine how fast the oil was produced. 

Secondary flooding process resulted in early water breakthrough (BT), but with continual oil production. This concurred with typical behavior of water flooding neutral-wet rocks [39] and it indicated the effectiveness of the ageing process. Additionally, it proved the ability of PSiNPs to improve microscopic sweep efficiency of water flood. The PVs injected at the BT point for nanofluid systems varied from 0.33 to 0.46 compared to 0.33 PV achieved by water. The small variations between water and nanofluid BT points may be due to the similarity in viscosity of water and nanofluid. Furthermore, it suggested that the PSiNPs were easily propagating through the cores plugs with little retention or no severe damage to core permeability. This was probably a manifestation of the role the polymer coating on the NPs surface played to providing steric repulsive forces between the particles, while enhancing the particle migration through Berea sandstone. The average oil recoveries at BT varied from 42% to 49% OOIP due to nanofluid flood compared to 40% achieved by water flood. The nanofluid NF3 had the highest average oil recovery of 49% OOIP at the BT, but the highest ultimate recovery was achieved with NF5. The oil recovery of NF5 correlated very well with the reduction of the IFT between oil and water, showing its greater contribution on EOR effect than other samples. The number of PVs injected for the nanofluid system to reach ultimate recovery varied from 5 to 10 PVs. This is a relatively large number of PVs of nanofluid injected. It may appear economically impracticable if directly compared to field-scale injection, as the cost factor for NPs will increase. However, at core scale, large pore volumes must be injected to produce comparable and realistic results for a field-scale applications [40]. The number of PVs injected in our experiments were also affected by initial core wettability and the ability of nanofluid to sweep oil.

The WF applied after nanofluid flooding incremented oil recovery in the range of 0.66% to 2.15% OOIP. It is possible that part of this oil has accumulated at the core outlet and was produced due to the increased flowrate. Increasing the flow-rate helps to negate the capillary end-effects, thus reducing the oil saturation at the core outlet. The low oil recovery by {WF} even with increased rate suggest that the capillary end-effects were small with nanofluid flood, which is consistent with observations made by [41] for neutral-wet systems.

In tertiary recovery mode, the PSiNP-based nanofluids could mobilize to some extent the residual oil. However, no significant oil recovery occurred during the first pore volumes of nanofluid injected. That implied that the tertiary oil recoveries of nanofluids were affected by the early stages of oil production and the mechanism entrapping the oil. Therefore, in order to mobilize this oil, it required the injections one or more PVs and allow for an extended physico-chemical interactions between the NPs and the rock system. The main oil bank was observed after large PVs of nanofluids were injected. However, most of the production was occurring in the form of disconnected oil drops. The late occurrence of oil due to nanofluid can be explained by physicochemical interactions between the {NP}s and the rock system [42,43]. The ultimate incremental oil recoveries of nanofluid varied from 2.6% to 5.2% of OOIP. The NF3 and NF4 showed superior behavior with displacement efficiency ranging from 6.3% to 11.8%, despite the increased pressure with their injection. Overall results showed that PSiNPs are promising EOR-agents in neutral-wet rocks.

### 4.2. Evaluation of Oil Recovery

Previous studies have shown the potential of silica NPs for increasing oil recovery. In some studies, the increased oil recovery ranges from 5% to 18% with silica-base nanofluids in the secondary oil recovery mode [12,20,21,22,44], whereas the incremental oil recovery of silica nanofluid varied from null up to 15% of OOIP after WF [6,7,13,19,20,45,46]. Most of these studies are carried out with NPs suspended in aqueous solution with low ionic strength (Na${}^{+}$ is the only cation in the solution), except Hendraningrat and Torster [19], who evaluated the effect of silica NPs in aqueous solution with divalent cations (e.g., Ca${}^{2+}$, Mg${}^{2+}$), and Ogolo et al. [12], who used non-aqueous solution. The concentration of NPs in the dispersion was varied. The flooding experiments were performed using different porous media and the oil phase varied from mineral oil to heavy oil. The oil, brine and rock composition affect the NPs interaction with the rock system. Thus, the parameters involved in the evaluation of silica or other NPs produced or will produce different results. This variability of flooding experiments makes it difficult to directly compare the effectiveness of the silica NPs for enhancing oil recovery. However, all flooding experiments may reveal a possible application of silica NPs for EOR under a variety of conditions that can be found in an oil field.

Comparing secondary oil recoveries from different nanofluid systems shown in Table 5, we see that sample NF5 had extremely increased oil recovery by 15.5% points, from 56% OOIP with WF to 71.5% OOIP with NF5 flood. The results suggest the use of NF5 for secondary oil recovery compared to the other nanofluids that increased oil recovery by a factor of 3–5.5% points. However, the choice for the EOR application of the nanofluids may be fully tied to the oil prices and the acquisition price of the nanomaterials, since in most cases WF would be preferred due to its low price. The oil recoveries provided in Table 5 and Table 6 show that secondary nanofluid flooding increased oil recovery by 3–15.5%, while after WF the nanofluids incremented oil recovery from 2.6% to 5.2% of OOIP. At first glance, these results may favor the use of the nanofluids as secondary EOR-agents. However, due to feasibility application of WF from start of oil recovery, nanofluid can be used as tertiary EOR-agent to exploit its benefit [20].

The nanoparticles studied in this work are still research and development products. The previous production line of these nanomaterials were tested by Aurand [37] in oil-wet cores. The nanomaterials were suspended in SSW with similar composition as used in this work at a concentration of 0.05 wt %. The author reported an additional oil recovery less than 3% OOIP and produced recommendations for further developments of the NPs. In her study, the oil-wet condition of the cores was achieved by ageing the cores in crude oil without water films. The NPs were re-improved by the manufacturer; the EOR evaluation procedure of the NPs was also improved in the current work. As noted above, an additional oil recovery after WF ranging 2.6–5.2% of OOIP was achieved with the improved NPs in this work, which shows a positive effect of the new particles. However, it is still believed that the surface chemistry of the nanoparticles need to be optimized.

### 4.3. EOR-Mechanisms of the Polymer-Coated Silica Nanoparticles

#### 4.3.1. Effect of PSiNPs on Interfacial Tension

In nanofluid flooding, the IFT between the flowing aqueous phase and oil phase can be reduced with NPs. Accordingly, the capillary forces preventing the oil from moving are decreased, and the oil recovery can be improved. This is due to the ultra-small NP size and the large surface area that allow the NPs to adsorb to the oil–water interface, even in the tiny pore spaces of a reservoir. The reduction of capillary forces promotes deformation and breaking of large oil droplets into small ones that can flow easily through the pore throats toward the production wells [7,47]. The effectiveness of NPs to mobilize the oil blobs can be evaluated by a balance between the viscous forces (mobilizing the oil) and capillary forces (trapping the oil) and it is characterized by capillary number ($N_{c}$):(2)$$N_{c}=\frac{\mu \nu }{\gamma }$$ where μ and ν represent the viscosity and velocity of the displacing fluid, respectively. γ is the IFT between the displacing and the displaced fluid.

Many studies have shown that the capillary number should be increased to >10${}^{-5}$ [5,48] so that the capillary trapped oil blobs can be produced. The IFT between SSW and oil measured in this work was 10.6 mN/m. The calculated capillary number was in the order of 10${}^{-6}$. The IFT values between nanofluids and oil reported in Figure 6 produced capillary numbers in the same order of magnitude as that of SSW and oil. Therefore, the IFT reduction cannot be a sufficient parameter to explain the oil recovery in this work. Such results are in agreement with other studies [25,26,33]. The modest reduction of IFT with nanofluids suggested that the PSiNPs were unable to significantly enhance the interaction between the immiscible oil and water phases, i.e., the PSiNPs were not surface active enough to dramatically reduce the IFT. A reason for this could be that the repulsive interactions between the particles surface and the liquid surface created a potential barrier that prevented the PSiNPs from adsorbing to the oil–water interface [49,50], even when the adsorption energy was favorable [50]. However, the modest adsorption of PSiNPs to the oil–water interface still plays a role for stabilization of emulsions [25]. Further studies are needed to investigate favorable conditions for adsorption of PSiNPs to the oil–water interface, so that recommendations can be made for a proper design and modification of NPs surface for better oil recovery.

#### 4.3.2. Effect of PSiNPs on Differential Pressure

The differential pressure (dP) was recorded during the flooding experiments using a Keller PD-33X with range of 0–3 bar instrument as a function of PVs injected. Two patterns of dP were identified, the decrease and increase of pressure with PSiNPs injection compared to “pure” WF pressure. It was observed that the dP decreased with the injection of nanofluids with small NPs size (NF1, NF2 and NF5) (Figure 4 and Figure 5a), in contrast to nanofluids with large size (NF3 and NF4), as illustrated in Figure 5b that increased the pressure. The results show an agreement with permeability results measured on cores after nanofluid flooding. That is, the larger was the NP size, the greater was the permeability reduction and vice versa, which is consistent with observations made by Adil et al. [42] and Hendraningrat et al. [16] in their experimental works. Nevertheless, no clear correlation was found between the dP and oil recovery with the initial NP size in this work.

When samples NF1, NF2 and NF5 were injected from start (at S${}_{wir}$), the dP increased until it reached nanofluid BT. Thereafter, the pressure declined and gradually stabilized. The pressure increased probably due to the confinement of NPs and blockage of small pores at the core inlet when the NPs attempted to establish favorable displacement paths through the randomly distributed core pore spaces. After the BT of nanofluid, the pressure dropped and remained relatively lower than water injection pressure. This indicated that the NPs were weakly adsorbed or confined at the core entrance. As the injection progressed, the NPs were transported into the core and likely produced without significantly clogging the porous spaces. The continuous oil production after nanofluid BT demonstrated the ability of NPs to improve microscopic sweep efficiency while lowering the injection pressure [27]. The mechanism could be the emulsification of oil into the water phase or formation of in-situ oil-in-water emulsions [42] with nanofluid injection. The emulsification of oil is also supported by low IFT results achieved, in particular, by samples NF1, NF2 and NF5 (i.e., >50%). It is most likely that the fraction of adsorbed PSiNPs to the oil–water stabilized the emulsions. El-Diasty [6], ShamsiJazeyi et al. [25] and Zhang et al. [43] reported that polymer-coated silica nanoparticles, such as used in this work, can produce stable emulsions that can travel through the cores with low retention resulting in improved oil recovery.

The pressure behaved differently after the injection of samples NF3 and NF4 compared to previous samples (NF1, NF2 and NF5) under the same flooding conditions. Figure 4a and Figure 5b show that the dP increased and it was higher than that produced by water injection. The observed variations in dP behavior, for repeated tests, can be explained by variations in core properties and initial paths established during early stage of water injection. The pressure analysis suggested that some NPs were being log-jammed throughout the flooding process and some pore spaces were blocked. It is important to note that samples NF3 and NF4 had the largest particle size and contained aluminum in their composition. We hypothesize that the polymers were weakly attached to the particle surface and the aluminum could interact with other components of the rock system to form sulfates. The sulfates could lead to the formation of precipitates that could block the porous spaces and increase the injection pressure. However, the maximum dP due to pore blocking was nearly 100 mbar, which is insufficient under the experimental conditions to produce desired capillary number to contribute significantly for oil recovery.

#### 4.3.3. Effect of PSiNPs on Rock Wettability

To investigate the effect of the polymer-coated silica nanopaticles (PSiNPs) on EOR during core flooding experiments, we explored the rate of water imbibition and Amott–Harvey wettability indexes (WIs). Figure 7 and Figure A3 (in Appendix B) show the results of spontaneous water imbibition (SI). The cores showed an improved water capillary intake and rapid oil recovery within the two day-test. These results were similar for all tests. It was observed that soaking the cores in nanofluids had no significant effect on wettability compared to the non-soaked cores. In both cases, the capillary pressure of the systems became positive with nanofluids, creating favorable conditions for water spontaneous imbibition. Amott–Harvey test was conducted to confirm the SI results. The measured WIs on cores after secondary and tertiary injection processes showed similar trend. Figure 9 shows only the WIs results measured on cores after tertiary nanofluid flood. The average WI was dramatically increased from −0.1, measured on aged cores in Section 3.1, to 0.58 (for soaked cores) and 0.61 (for non-soaked cores). According to the categorization of wettability based on WIs given by Anderson [38], the WIs indicate results of water-wet, which agrees with SI results. The variations in WIs may be related to core properties and accuracy of measurements.

To elucidate the mechanism of wettability alteration of PSiNPs, further studies are needed. It is more likely that the change in wettability occurred due to adsorption of NPs involving physicochemical interactions during the nanofluid injection process. Since the PSiNPs were hydrophilic and the rock surface was neutral-wet, the wettability alteration occurred through the disjoning pressure mechanism on water-wet pores, as hypothesized by Wasan and Nikolov [32].

Unlike in oil-wet pores, the wetting mechanism is complex. However, it is believed that the surface interactions between the oil and the rock was decreased with the injected NPs. The weakly attached oil films on the rock surface may have been disrupted allowing the NPs to adsorb or being confined in the pores. The large amount of particles injected along with the prolonged injection period may have played an important role in increasing the adsorption of NPs on the surface of the rock, thus developing new surface roughness that altered the wetting properties of the rock surface. These findings highlighted the potential of the polymer-coated silica nanoparticles (PSiNPs) for oil recovery. The wettability alteration from neutral-wet to water-wet condition is likely the main EOR mechanism of the studied nanofluids.

#### 4.3.4. Effect of PSiNPs Permeability

After core flooding and wettability tests, the cores were cleaned and dried. Then, the core absolute permeability was determined. The results are summarized in Table 7. The NF3 and NF4 showed the largest absolute permeability reduction ranging from 23% to 35%. The results correlated positively well with the initial size of the particles (i.e., higher permeability reduction with higher particle size). Contrasting permeability results were observed on cores injected with NF1, NF2 and NF5. The permeability of the cores was slightly reduced or improved. Accordingly, no correlation could be established with their primary size.

It is possible that the results were affected by the use of centrifuge to perform forced displacements. The rock fines may be released and the NPs desorbed from the rock surface due to high centrifugal forces, which can increase the rock permeability to some extent. In fact, some sand grains were observed glued to the walls of the glass jars when measuring the amount of fluids produced by centrifuge. This indicated a possible detachment of sand grains from the rock surface. The extent to which the change in rock permeability would be if the centrifuge were not used is uncertain. However, the experimental results indicate negligible reduction of permeability with nanofluids.

#### 4.3.5. Nanofluid Stability

The stability of the polymer-coated silica nanoparticles (PSiNPs) were studied using particle size distribution and zeta potential measured with dynamic light scattering method, and sedimentation test for four months. Malvern Zetasizer instruments were used to obtain the dominant size of the particles and their zeta potential. The measured particle size is reported on average of three measurements. The average sizes are given in Table 8. The primary particle size of the particles dispersed in distilled water (as provided by manufacturer) was decreased when the solutions (concentrated solution) of NPs were diluted with SSW to 0.1 wt %, except for NF2, whose particle size slightly increased. This suggested that there was no formation of aggregates/agglomerates in the SSW solution. The increased particle size for sample NF2 may be an indication of growth in initial particle size. However, assuming that an increase or decrease in primary particle size by 50% would provide a reasonable conclusion regarding the stability of the particles, then the results presented here suggest that all nanofluids were stable under the experimental conditions. This was a positive effect of the polymer coating on the particle surface, since the SSW would increase the tendency for agglomeration of the particles due to the increased ionic strength. Measurement of zeta potential of the NPs gave unreliable results for unknown reasons. Sedimentation tests were then conducted to confirm the particle size measurements. After the four-month period, no visual aggregates/agglomerates were observed at ambient conditions, which validated the characterization of the particle results. The formation of aggregates/agglomerates suggested by increased dP across the core for NF3 and NF4 can probably be attributed to the mechanical entrapment of PSiNPs within the cores during the injection process and the properties of the cores.

## 5. Conclusions

This work investigated the effect of polymer-coated silica (PSiNPs) nanoparticles as additives to seawater injection for enhancing oil recovery in neutral-wet Berea sandstone rocks. The main EOR mechanisms of NPs were also studied. Based on the experimental results, the following conclusions were obtained:Stability analysis revealed that the polymer-coated silica nanoparticles were stable in synthetic seawater under the experimental conditions for more than four months.Core flooding experiments revealed that oil is enhanced with polymer-coated silica nanoparticles in neutral-wet Berea sandstone reservoirs. The nanofluids increased oil recovery up to a factor of 16% in the secondary recovery scheme. In tertiary recovery mode, the PSiNPs improved oil recovery from 2.6% to 5.2% OOIP.Interfacial tension measurements showed that the tension between crude oil and seawater was reduced by PSiNPs, but the reduction cannot fully explain the oil recovery effect.Nanofluids (NF1, NF2 and NF5) improved microscopic sweep efficiency through a combination of IFT reduction, formation of in-situ emulsions and wettability alteration.Nanofluids (NF3 and NF4) improved microscopic sweep efficiency through a synergistic effect of IFT reduction, pore blocking and flow diversion, and wettability alteration.All nanofluid systems altered the wettability from neutral-wet to water-wet condition, which is likely the main oil recovery mechanism of the nanoparticles.

Further work is ongoing using water-wet and neutral-wet core plugs, to evaluate the EOR potential of the polymer-coated silica nanoparticles at elevated temperature. Based on the current findings, additional studies will be conducted to address thoroughly the oil recovery mechanisms.

## Figures and Tables

**Figure 1 nanomaterials-09-00822-f001:**
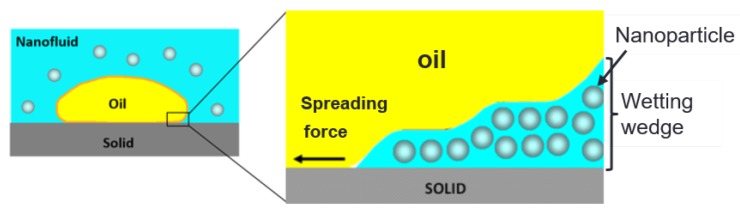
Nanoparticles structuring in the wetting wedge and oil displacement driven by structural disjoining pressure gradient at the wedge vertex [32].

**Figure 2 nanomaterials-09-00822-f002:**
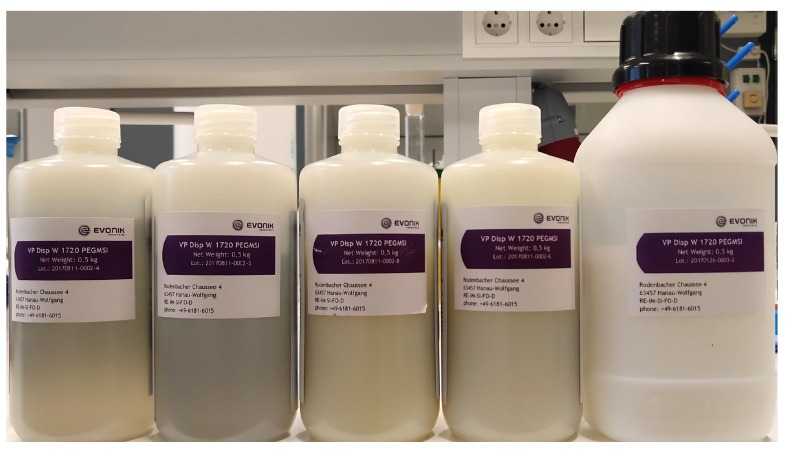
Concentrated solutions of hydrophilic silica nanoparticles as received from Evonik Industries; from left to the right: NF1, NF2, NF3, NF4 and NF5.

**Figure 3 nanomaterials-09-00822-f003:**
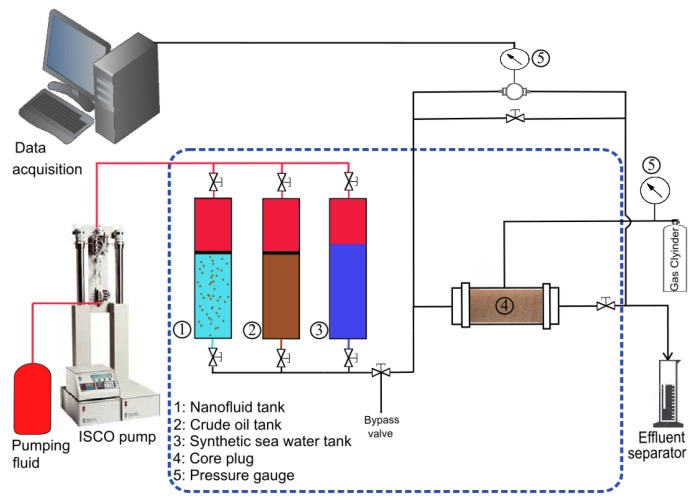
Schematic diagram of core flooding experiment.

**Figure 4 nanomaterials-09-00822-f004:**
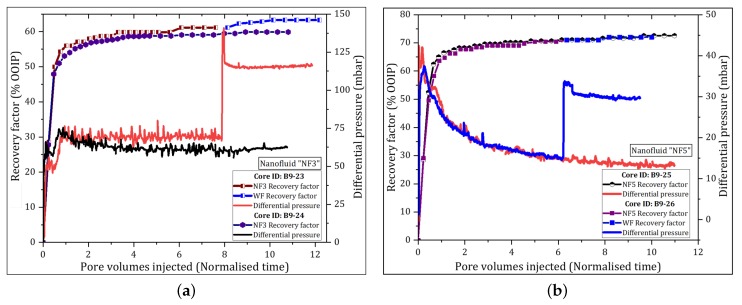
Secondary oil recoveries and dP versus PVs recorded throughout flooding tests. Two tests were conducted for each nanofluid type. WF was conducted only in replicate test at flow rate of 0.5 mL/min: (**a**) NF3 (dP = 65 mbar); and (**b**) NF5 (dP = 14 mbar).

**Figure 5 nanomaterials-09-00822-f005:**
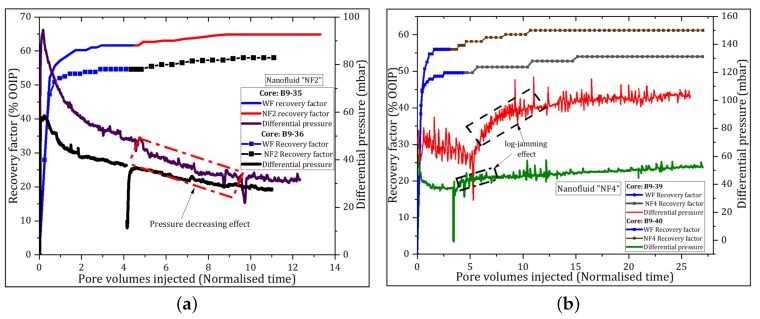
Tertiary oil recovery factors and dP versus PV recorded during the nanofluid flooding. Two parallel tests were conducted for each nanofluid type: (**a**) NF2; and (**b**) NF4.

**Figure 6 nanomaterials-09-00822-f006:**
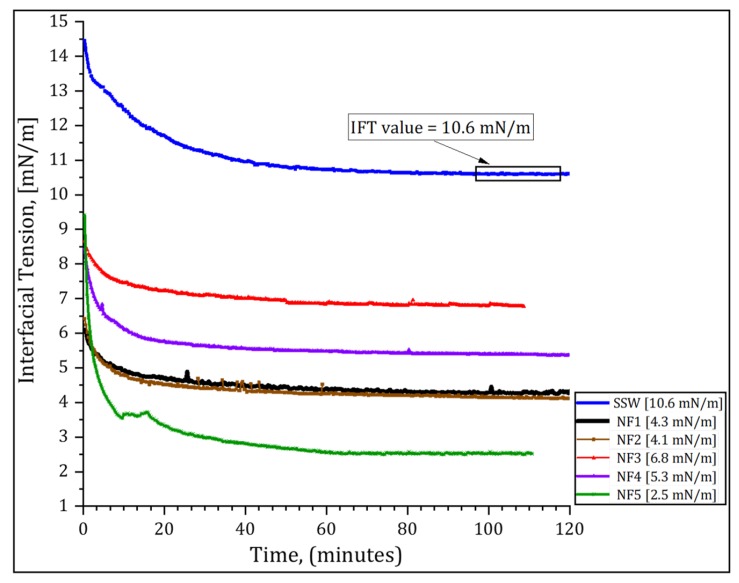
Interfacial tension values between crude oil and aqueous solutions of polymer-coated silica nanoparticles.

**Figure 7 nanomaterials-09-00822-f007:**
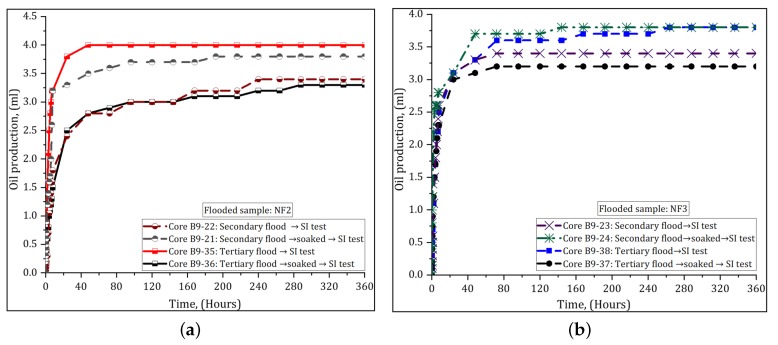
Spontaneous imbibition (SI) behavior curves started at S${}_{wir}$: (**a**) core plugs flooded with NF2; and (**b**) core plugs flooded with NF3.

**Figure 8 nanomaterials-09-00822-f008:**
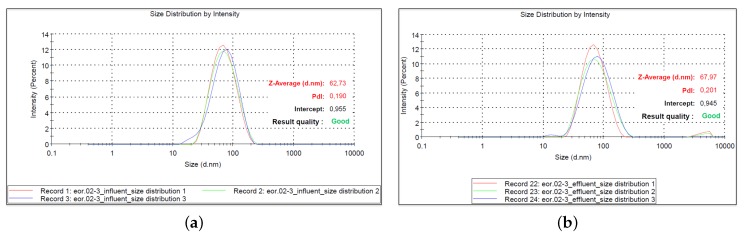
Nanoparticle size distribution curves for sample NF1 at 0.1 wt %: (**a**) influent; and (**b**) effluent.

**Figure 9 nanomaterials-09-00822-f009:**
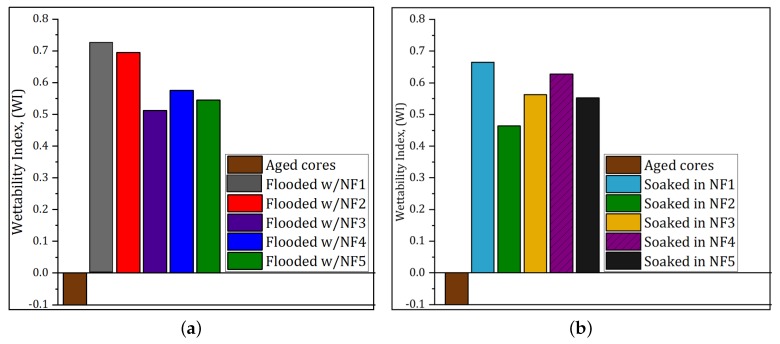
Amott–Harvey wettability indexs of reference aged cores, and that measured on cores after: (**a**) nanofluid injection; and (**b**) nanofluid injection plus soaking in the same nanofluid.

**Table 1 nanomaterials-09-00822-t001:** Properties of the nanoparticles suspended in distilled water.

Sample	Basis	Modification	Concentration, wt %	Size (nm)
NF1	SiO${}_{2}$ (sol-gel-cationic)	Polymer	38.6	107
NF2	SiO${}_{2}$ (sol-gel-anionic)	Polymer	26.0	32
NF3	SiO${}_{2}$/Al${}_{2}$O${}_{3}$/MOX	Polymer	21.6	218
NF4	SiO${}_{2}$/Al${}_{2}$O${}_{3}$/MOX	Polymer	25.5	145
NF5	SiO${}_{2}$/Al${}_{2}$O${}_{3}$/MOX	Polymer + additives	27.0	112

**Table 2 nanomaterials-09-00822-t002:** Fluid properties measured at room temperature (20 ${}^{\circ }$C).

Fluid	Density (g/cm${}^{3}$)	Viscosity (cP)	pH
SSW	1.024	1.025	7.97
NF	1.023–1.028	1.022–1.057	7.73–8.11

**Table 3 nanomaterials-09-00822-t003:** Crude oil properties measured at 21.5 ${}^{\circ }$C and the SARA analysis (normalized wt %).

Crude Oil	ρ (g/cm${}^{3}$)	μ (cP)	Saturates (%)	Aromates (%)	Resins (%)	Asphaltenes (%)
A	0.904	52.5	66.21	25.78	7.69	0.32
B	0.886	34.0	71.57	20.81	7.44	0.18

**Table 4 nanomaterials-09-00822-t004:** Amott–Harvey wettability indexes measured on aged cores.

Core	V${}_{o1}$	V${}_{o2}$	V${}_{w1}$	V${}_{w2}$	I${}_{w}$	I${}_{o}$	WI
W1	0.0	4.5	0.3	3.0	0.00	0.09	−0.09
W2	0.1	6.0	0.5	4.0	0.01	0.11	−0.10
W3	0.0	3.0	0.3	4.0	0.00	0.07	−0.07
W4	0.0	6.5	0.5	4.0	0.00	0.11	−0.11
W5	0.1	6.5	0.5	4.0	0.01	0.11	−0.10
W6	0.0	4.0	0.4	2.0	0.00	0.17	−0.17

**Table 5 nanomaterials-09-00822-t005:** Oil recovery factors (water and nanofluid flooding), expressed as percent of OOIP, and residual oil saturation achieved at the end of core flooding in neutral-wet cores.

Injected			Nanofluid Flood					Water Flood		Total RF
Fluid	PV@BT	RF@BT	PV@max.RF	RF	S${}_{\mathrm{or}}$	$\bar{\mathrm{RF}}$	% points	RF	S${}_{\mathrm{or}}$	
Water ^1^	0.33	40.22	2.93	-	-	-	-	56.05	35.57	56.05
NF1	0.39	44.44	7.33	62.87	32.55	61.50	5.5	0.66	31.97	63.53
	0.35	40.52	7.05	60.13	35.83			-	-	60.13
NF2	0.33	40.79	4.42	59.50	35.86	59.30	3.3	1.42	34.58	60.92
	0.34	42.31	9.22	59.10	36.78			-	-	59.10
NF3	0.43	50.00	5.58	61.14	34.52	60.52	4.5	2.15	32.19	63.29
	0.36	47.22	8.93	59.89	33.92			-	-	59.89
NF5	0.34	41.89	5.00	70.41	24.89	71.50	15.5	1.62	23.52	72.03
	0.37	47.29	9.66	72.57	22.29			-	-	72.57

${}^{1}$ Average values of eight water flooding tests (see Table 6).

**Table 6 nanomaterials-09-00822-t006:** Oil recovery factors (water and nanofluid flooding), expressed as percent of OOIP, and $S_{or}$ achieved at the end of core flooding in neutral-wet cores.

	Water Flood		Nanofluid Flood					
Sample	RF	S${}_{\mathrm{or}}$ (%)	PV@1st Oil	RF@1PV	RF	S${}_{\mathrm{or}2}$ (%)	E${}_{D}$ (%)	Total RF
NF1 ^2^	58.86	36.83	0.9	0.71	2.71	34.39	6.6	61.57
	51.80	40.39	-	-		–	-	-
NF2	61.33	35.72	0.9	1.1	3.33	32.64	8.6	64.67
	54.70	40.96	1.3	0.00	3.36	37.98	7.3	58.03
NF3	59.29	32.26	1.1	0.00	3.29	29.66	8.1	62.57
	58.05	35.10	1.6	0.00	4.17	31.62	9.9	62.22
NF4	49.59	38.16	1.5	0.00	4.44	34.80	8.8	54.03
	55.97	37.41	1.2	0.00	5.21	32.99	11.8	61.17
NF5	36.38	1.1	0.00	2.85	33.94	6.7	60.14
	59.14	31.57	1.8	0.00	2.57	29.58	6.3	61.71
Average	56.05	36.50	-	-	3.5	33.06	-	60.00

^2^ The replicate test failed.

**Table 7 nanomaterials-09-00822-t007:** Permeability of the cores measured before and after secondary (left) and tertiary (right) nanofluid flooding.

Core	Injected Sample	Permeability (mD)		% Difference	Core	Permeability (mD)		% Difference
		Before	After			Before	After	
B9-19	NF1	321	285	−13	B9-33	228	282	19
B9-20		391	388	−1	B9-34	-	-	-
B9-21	NF2	238	207	−15	B9-35	238	207	−15
B9-22		368	343	−7	B9-36	256	284	10
B9-23	NF3	248	195	−27	B9-37	248	195	−35
B9-24		304	236	−29	B9-38	270	220	−23
	NF4	-	-	-	B9-39	233	190	−18
		-	-	-	B9-40	234	200	−17
B9-25	NF5	347	299	−16	B9-41	230	245	6
B9-26		301	281	−7	B9-42	247	220	−12

**Table 8 nanomaterials-09-00822-t008:** The primary NP diameter in distilled water was provided by the manufacturer. The columns under influent, effluent (1) and effluent (2) report the average size of the NPs before injection, after secondary and tertiary nanofluid flooding, respectively.

Sample	Average Nanoparticle Diameter (nm)				
	Provided	Influent	% Difference	Effluent (1)	Effluent (2)
NF1	107	63	−41	61	68
NF2	32	38	19	50	39
NF3	218	154	−29	159	161
NF4	145	135	−7	127	132
NF5	112	109	−3	109	107

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
