# Peer review of "Experimental Investigation of Polymer-Coated Silica Nanoparticles for Enhanced Oil Recovery"

_nanomaterials, 2019, doi:10.3390/nano9060822_

Reviewer 1 Report

Authors described synthesis and use of different types of 'polymer coated silica nanoparticles' as additives to the injection seawater and its probables role in EOR at either secondary or tertiary stage in core-flood experiments. They also reported other supportive experiments (to understand and explain the role of such system during EOR) such as Amott tests, IFT, particle size distribution etc. All the methods followed are standard, well written, and quite informative. However, there are few  suggestions for missing key-information and other minor changes, as mentioned below, which needs to be addressed in the revised manuscript:

- L106-108: 'The surface of the NPs was modified with polymer chains to render them stable in the injection synthetic seawater (SSW). These types of NPs are referred to as polymer-coated silica nanoparticle (PSiNP) hereafter.' - You should mention here, what type of polymer/chains you used to coat SiNPs? What was the molecular weight/ionic nature? Such information is needed to give a clear picture to the reader for repeatability and other applications if possible.

- Table 1: Similarly here also it is not clear when you mention 'Polymer'/'Polymer+additives'?  What type of polymer and additives were those? You should mention here/or in text.

- L139-149: You should mention here how long these experiments were carried out for each steps (with respect to time duration for such Amott tests).

- L238-239: Would rather suggest to add this figure here, which showed better oil recovery and reduction in dP!

- L241-242: 'This included sample NF02-8, because it was hypothesized would give good results.' - How/why did you hypothesized it to work better? Was it because of the size/polymer?

- L337-338: Yes, indeed it is quite high PV to be injected, which is not economically practicable at field scale! As the cost factor for NPs/Polymer will also be in the picture. 

- It would be good to see some more comparison with SiNPs or such combinations reported by others with your results, in the discussion part. Such as, how different or better are the results.

- References: Why reference numbers start from 43! Both in-text and at the end as well! Seems like some typesetting mistake. Please cross-check and correct.

Author Response

Dear reviewer,

Thank you very much for your comments. Please find the attached coverletter.

I hope i have addressed the comments and suggestions thoroughly. I am looking forward to hearing from you. 

best regards

Alberto

Reviewer 2 Report

The author present a study on the influence of polymer-coated silica nanoparticles for oil recovery. The experiments and methods are explained in detail.

With "polymer-coated silica nanoparticles" in the title it sounds like an interesting study suited for Nanomaterials. Unfortunately the information given about the different nanoparticles e.g. about the nature of the polymer coating, is limited. I am aware that the company might not disclose all the information about their products, but I cannot understand why the trade names of the products of  Evonik industries are not given. Of course I can look into the product overview of the company and try to find some nanoparticles matching the descriptions, but this is ambiguous. How one should be able to reproduce the experiments?

Instead the study uses unnecessary complicated code for samples and experiments (e.g. NF02-3, NF02-4). Why not keep it simple with NF1, NF2... or even A, B,.. ?

Overall a lot of abbreviations are used in the text. Although all abbreviations are explained, the readability suffers.

The graphs should be improved including color choices (different shades of green in figure 3a) and the "fancy" patterns of the bar charts in figure 5 and 8.

Table 8 claims to give information about the size distributions but only provides single numbers without any distribution. Is it expected that the reader extracts this information from figure 7 and the appendix himself? Also the difference to the particle size information from the manufacturer should be discussed (What is the experimental source for the data from the manufacturer?).

Author Response

Dear reviewer,

Thank you very much for your comments. Please find the attached coverletter. I hope i have addressed the comments and suggestions thoroughly. I am looking forward to hearing from you. 

best regards

Alberto

Round  2

Reviewer 1 Report

Authors amended the manuscript with all suggestions, and added further information. Original manuscript was very informative, revised version reads even better. 

Reviewer 2 Report

The manuscript has been improved and I have no further comments to the revised version.